# Continuous Protein Supplementation Reduces Acute Exercise-Induced Stress Markers in Athletes Performing Marathon

**DOI:** 10.3390/nu13092929

**Published:** 2021-08-24

**Authors:** Martin Röhling, David McCarthy, Aloys Berg

**Affiliations:** 1West-German Centre of Diabetes and Health, Düsseldorf Catholic Hospital Group, 40591 Dusseldorf, Germany; martin.roehling@vkkd-kliniken.de; 2Public Health Nutrition Research Group, London Metropolitan University, London N7 8DB, UK; d.mccarthy@londonmet.ac.uk; 3Faculty of Medicine, University of Freiburg, 79117 Freiburg, Germany

**Keywords:** protein-rich diet, endurance exercise, muscle stress, inflammation

## Abstract

The aim of this study was to determine the changes in endurance performance and metabolic, hormonal, and inflammatory markers induced by endurance stress (marathon race) in a combined strategy of training and dietary protein supplementation. The study was designed as a randomised controlled trial consisting of regular endurance training without and with a daily intake of a soy protein-based supplement over a three-month period in 2 × 15 (10 males and 5 females per group) endurance-trained adults. Body composition (body mass, BMI, and fat mass) was determined, and physical fitness was measured by treadmill ergometry at baseline and after 3 months of intervention; changes in exercise-induced stress and inflammatory markers (CK, myoglobin, interleukin-6, cortisol, and leukocytes) were also determined before and after a marathon competition; eating behaviour was documented before and after intervention by a three-day diet diary. Although no significant influence on endurance performance was observed, the protein supplementation regime reduced the exercise-induced muscle stress response. Furthermore, a protein intake of ≥20% of total energy intake led to a lower-level stress reaction after the marathon race. In conclusion, supplementary protein intake may influence exercise-induced muscle stress reactions by changing cellular metabolism and inflammatory pathways.

## 1. Introduction

It is well accepted that nutrition is a limiting factor in human performance and that athletes should ensure that they meet their daily energy needs to balance their requirements for energy expenditure [1]. Independent of total energy intake, the specific timing and macronutrient composition of energy intake throughout the day has the potential to modulate the body’s adaptation to exercise [2]. Furthermore, macronutrient composition should be adapted to the regularly performed training or sport modality since different sports have different energetic and nutritional requirements [3]. 

With respect to dietary protein, the supply of essential amino acids is critically linked to muscle protein synthesis and can promote skeletal muscle hypertrophy in response to chronic resistance training [4]. However, evidence linking protein intake to a benefit in endurance sports has not been clearly established [5]. Authors have suggested that protein intake during or immediately after exercise may induce an ergogenic effect on endurance performance, preferably in combination with the intake of carbohydrates as the first-choice energy supply [2,6,7,8]. Nevertheless, there remains no consensus on whether the metabolic and systemic effects of proteins is simply a consequence of providing extra energy or rather, may be a specific benefit regarding the protein source as well as to the intake of biological active peptides, which are also increasingly absorbed via the increased or targeted supply of proteins [9,10,11].

Therefore, we performed a randomised controlled trial (RCT) involving a group of experienced athletes to evaluate the effects of a combination of endurance training and a regular intake of a protein-rich dietary supplement over a period of 3 months in preparation for a marathon race. These athletes also partook in a local marathon at the end of the intervention. The main aim of the study was to investigate whether the intake of the protein supplement influences endurance performance and whether its supplementation reduces exercise-induced muscular or systemic stress reactions.

## 2. Materials and Methods

### 2.1. Study Design

The study was designed as an experimental RCT to determine the effects of a protein-rich supplementation regime across a 3-month training period on the endurance performance and training adaptation, ultimately evaluated with a marathon race at the end of the intervention compared to a control group. Participants study visits were conducted at baseline (termed “a”) and at the end of the 3-month training intervention (termed “b”) as well as before (termed “c”) and 1 h after (termed “d”) the marathon race at the end of intervention. The period between visits “b” and “c” lasted approximately 7 days, but the athletes had to continue with the protein supplementation regimen until the marathon race day. After the “a" visit, study participants were randomly assigned either to the intervention or control group. 

### 2.2. Participants

After contacting and advertising the study protocol to regional sports clubs, 30 out of 52 applicants met the criteria for randomisation (defined as experienced endurance runners) to either a verum (V; *n* = 15, 10 males, 5 females) or a control group (C; *n* = 15, 10 males, 5 females). Participants had to fulfil the following inclusion criteria: to be clinically healthy endurance-trained adults, aged between 18–60 years, and a body mass index (BMI) between 18–25 kg/m^2^. Additional inclusion criteria were: constant roadwork training (ca. 3 times per week of 25–30 km endurance training), non-use of other nutritional or ergogenic supplements, no blood lipid- or glucose-lowering therapy within the preceding 12 months, being non-smoker, alcohol consumption <40 g/d, and a relative maximal oxygen consumption (VO_2max_) between 45–65 mL/kg/min. The study protocol was registered and approved by the Ethical Commission of the University of Freiburg (reg. no. 231/12). The study was conducted in accordance with the ethical standards established in the 1964 Declaration of Helsinki and its later amendments [12]. All participants were informed verbally and supported by a written participant information sheet, and signed consent to participate in the study was obtained.

### 2.3. Protein Supplementation

The supplement used (comparable with the commercially available product Almased-Vitalkost^®^; Almased-Wellness-GmbH, Bienenbüttel, Germany) is composed of soy protein, skimmed milk powder, and honey without additives. The protein content of this product is 53.3% (83% soy-protein-isolate, and 17% milk protein). Given the special marathon conditions, each 50 g portion was supplemented with 9 g palatinose, 30 mg carnitine, 10 mg coenzyme Q10, and 0.5 mg α-lipoic acid. One portion thereby equated to 60 g of powder dissolved in 300 mL water, providing 217 kcal, 27.2 g protein, 24.6 g CHO, and 1.0 g fat. Participants were instructed to drink one portion twice per day, in the morning before breakfast and in the afternoon or evening, with each intake always approximately at the same time of day and with a delay before ingestion of other meals to account for the low glycaemic load of the product.

### 2.4. Measures

The participants’ accordance with the inclusion criteria was confirmed in the initial comprehensive health screening before the intervention. Laboratory analysis of blood samples as well as the measurement of anthropometric parameters were concomitantly performed. Anthropometry comprised body height (BH), body mass (BM), and body mass index (BMI). BM was measured in light clothing to the nearest 0.1 kg, BH to the nearest 0.5 cm (Seca Stadiometer 274, Seca GmbH & Co. KG, Hamburg, Germany), and BMI was calculated with the formula: weight in kilogram divided by height in meters squared. Percentage body fat (FM%) was calculated using the Siri equation [13] with the log sum of skinfold thickness (SFT) at 4 measuring sides (triceps, biceps, subscapular, and suprailia). The SFT was measured using a skinfold calliper (Lange Skinfold Caliper, Beta Technology Inc., Noblesville, IN, USA). The laboratory analyses included selected metabolic variables for performance diagnostics (blood lactate concentration [14]), whole blood cell counts (including leukocytes [15]), selected blood indicators of muscle damage and systemic stress (creatine kinase (CK) [16], myoglobin [17] and interleukin-6 (IL-6) [18]), and the hormone cortisol [19]. Biochemical blood variables were determined by venous blood sampling. Leukocyte count, cortisol, and indicators of muscle damage and stress response were utilised in the calculation of a “stress score” (unpublished “in-house” method to determine overall exercise-induced stress response), which was expected to be noticeably increased after bouts of hard endurance exercise, such as a marathon race. For this purpose, the participants’ values for each of the variables were ranked in ascending order and categorised into three equally sized groups. For each parameter, the participants were assigned to one of the three categories, reaching a stress score of 1 for a relatively low parameter increase, 2 for a moderate increase, and 3 for a high increase. Hence, for each participant, a final total stress score was calculated ranging from a possible minimum of 5, showing low levels for all 5 variables, to a maximum score of 15. Mean stress scores were calculated for both study groups. The comprehensive health screening also included performance diagnostics to assess the training status of the participants and were conducted as a treadmill test using spiroergometric equipment (ZAN Messgeräte GmbH, Oberthulba, Germany) following the ZAN protocol of sequential 3 min steps with increasing running speed [14,20]. After each increment to the maximal possible load (running speed max), a capillary blood sample was taken from the earlobe for blood lactate concentration measurement as well as immediately after the termination of the test to obtain the maximal blood lactate concentration. These lactate values were used to calculate the individual anaerobic threshold (IAT) [21]. The running speeds at the described thresholds were transformed into the theoretical time needed individually to reach 1000 m (1000 m time at the IAT). Participants were tested 3 h after breakfast. Blood samples were also drawn prior to performance diagnostics. In addition, the participants were requested to avoid highly intensive physical activity and strength exercise on the previous day to prevent post-training alterations in the pre-test blood samples acquired. 

### 2.5. Training Phase

To estimate training compliance and comparability of training volume and exercise intensity between the groups, the participants were requested to complete training protocols, documenting weekly training distances and training time over the 12-week intervention phase. The entire training phase was supervised by experienced endurance trainers in both groups. Furthermore, participants were requested to complete a 3-day estimated nutrition protocol before and after the intervention period and were advised to maintain their typical eating and drinking patterns. The evaluation of the food records was performed using the EBISpro nutrition system (Stuttgart, Germany) [22].

### 2.6. Statistics

Prior to analysis, data were checked for normality distribution. Intragroup comparisons between baseline and post-intervention as well as before and after the marathon race were assessed by the Wilcoxon signed-rank test. Intergroup comparisons were analysed by the Mann–Whitney U test. Spearman’s rank correlation analysis was performed to determine the influence of macronutrient intake on muscle stress reactions. The variables “Diff. Myoglobin (µg/L)” and “Diff. CK (U/L)” were log transformed due to variance inhomogeneity and the analyses (correlation, regression analysis and *t*-test) were conducted with the transformed variables. SPSS 22.0 (SPSS Inc., Chicago, IL, USA) was used for the statistical analysis. All statistical tests were two-sided, and the level of significance was set at α = 0.05.

## 3. Results

From the 30 endurance athletes (verum group: *n* = 15, 10 males, 5 females; control group: *n* = 15, 10 males, 5 females) recruited, 23 completed the 3-month intervention period; no adverse event could be related to the given formula (verum group: 5 dropouts (respiratory tract infection: *n* = 1, orthopaedic problems: *n* = 2, personal reasons: *n* = 2); control group: 2 dropouts (orthopaedic problems: *n* = 1, personal reason: *n* = 1)). However, 21 of these 23 athletes were able to perform the marathon race over the complete distance (2 dropouts because of muscle cramps) (Figure 1).

Basic and anthropometric characteristics of the study groups are presented in Table 1; no significant differences were found in these variables in the intra- or inter-group comparisons. Most of the verum group participants achieved a daily protein intake greater than 20% of the total energy intake, based on the 3-day diet diaries indicating intervention compliance regarding the protein supplementation regime.

Energy (EI) and macronutrient intakes (expressed as a percentage of EI and relative to BW) in the tested study groups at the beginning and the end of the intervention are presented in Appendix A. The intervention did not influence the total energy intake in either group. However, the supplement intake resulted in a significant increase in the relative and absolute protein intake in the verum group and yielded a significantly higher protein intake in the intergroup comparisons at the end of the intervention. Given the unchanged total energy intake, the increased protein intake in the verum group was associated with a significant decrease in relative fat intake across the course of intervention. 

Performance and exercise-induced stress variables measured before and after the intervention are shown in Table 2. There were no significant differences in the inter-group comparisons for any of these variables. As a result of the regular training sessions, both running time and running distance accumulated over the 12 weeks of intervention in both groups to a comparable extent (verum group: 3.100 ± 1000 min, 540 ± 230 km; control group: 2.810 ± 570 min, 571 ± 96 km). Aerobic performance showed a slight improvement of approximately 3% in both groups. 

Acute exercise-induced stress markers before and 1 h after the marathon race are shown in Table 3. Twenty-one participants completed the marathon race and were included in the analysis. There was no difference in the marathon running time in the tested groups (verum: 3:54 h ± 32 min; control: 3:59 h ± 26 min). Both groups showed significant increases in all exercise-induced stress markers. There were trends toward higher calculated increments for all stress markers in the control group. A significant inter-group difference was found for serum myoglobin.

Regression analyses among the increases in the measured stress variables and calculated stress score showed that there was a negative association between the individual protein intake (expressed as a proportion of daily protein intake in the post-intervention phase) and the increase in stress markers (significant difference in IL-6, cortisol, and stress score; trend for difference in CK, myoglobin, and leukocytes) after the marathon race (Table 4). The regression curves showed that with a daily protein intake (PI) of less than 20% of the total energy intake, the increase in stress markers was visibly higher. This was statistically confirmed in a group comparison for the participants of the marathon in relation to their protein intake (PI) (*n* = 12: PI < 20% vs. *n* = 9: PI = >20%) (Table 4). The graphical representation based on the stress score (Figure 2) also showed that an increased protein intake of more than 20% was achieved primarily by participants in the verum group. In contrast to that of the protein intake, the regression analysis did not provide any evidence for a relation between fat and carbohydrate intake and the exercise-induced stress reaction.

## 4. Discussion

In the present RCT, intake of a protein-rich supplement during a 3-month training intervention led to a reduced acute stress response after an exhausting marathon race, which negatively correlated with the proportion of protein intake in relation to the daily energy intake. Furthermore, a threshold of approximately 20% of energy intake from protein appeared to be necessary to reduce the increase in stress markers to lower levels. Whilst protein supplementation can be beneficial for athletes in relation to protein synthesis and reductions in muscle damage and enhanced recovery of muscle function [23], recently published RCT’s have nevertheless demonstrated that acute (1–3 days prior of exhausting exercise) [24], and post-exercise [25,26] protein supplementation is not superior compared to other types of dietary supplementation (e.g., carbohydrate supplementation) in order to attenuate exercise-induced muscle damage or systemic inflammatory processes. However, chronic protein supplementation for at least 1 month prior to an exhausting bout of exercise [27], which is consistent with the present findings, has been shown to induce beneficial effects on markers of muscle damage and inflammation. Apart from the influence of the supplementation duration prior to strenuous exercise, there are additional factors (e.g., training status (untrained vs. trained) or modality (resistance vs. endurance exercise)) that may impact the effect of the protein supplementation [28]. In the case of training modality, both types, endurance and resistance exercise accompanied by protein supplementation, can support positive adaptive responses in protein synthesis and reductions in muscle damage and enhanced recovery of muscle function [17,28,29]. 

The threshold of at least 20% of energy intake from protein sources in the present study points towards a dose-response pattern for protein supplementation. This finding is consistent with the current opinion that active individuals attempting to optimize training adaptations need to increase their protein intake from a typical level of 0.8 g/kg/day to at least 1.4–1.6 g/kg/day [30]. In addition to the quantitative aspect, qualitative aspects of the protein source may also be important, e.g., in relation to the uptake of biologically active peptides, which may have supported biological functions in the present study [31,32,33]. Furthermore, with respect to the regulation of anti-oxidative and anti-inflammatory actions, the role of isoflavones (which are a well-characterised component of the soy protein supplement) must be considered, as the protein supplement contains an amount of biologically available isoflavonoids of approximately 1.5 mg per g of dried powder. This intake may be at a level sufficient to produce biologically significant increases in blood isoflavone levels and potentially confer metabolic and anti-oxidative benefits [34]. As the study participants showed comparable exercise performance and were well prepared for the marathon race, the differences could not be explained either by their adaptation to endurance exercise or by the intensity, duration, and mode of the exercise performed [15,35].

The protein supplementation regime affected neither the training-induced improvement in the performance markers measured after intervention nor the marathon running time between the tested groups. These findings can be explained by the adaptation to the regularly performed endurance training before the study, as all participants were professionally orientated runners [14,20]. Conversely, both groups were equally supervised by licenced athletic and endurance trainers throughout the entire training phase. Another explanation may be that the additionally supplemented ergogenic aids of carnitine, coenzyme Q10, and α-lipoic acid abolished possible inter-group differences. In this case, it must be considered that the amount of the additives consumed was only in the range of the assumed daily requirement and not at the level provided in food supplements available on the market with the claim of improving aerobic performance. Thus, the results of this study do not support the assumption of an ergogenic effect of protein supplementation and most ergogenic aids in trained individuals accustomed to endurance sports, but instead agree with the position statement of the ACSM regarding nutrition and athletic performance Furthermore, high carbohydrate intake in the preparation as well as during an endurance event remain as the best-documented factor for endurance performance [36,37,38,39].

The following strengths and limitations should be considered when interpreting the data. We were able to motivate 30 long distance running experienced participants to partake in the study [1,30]. However, based on the small sample size, any tendencies in outcomes observed between both groups did not reach statistical significance and can only be interpreted with caution as possible trends. In addition, it must be considered for the quality of the study because it was not conducted using a placebo-controlled design since it was impossible to produce a placebo supplementation formula without the main components but with a comparable taste and appearance as well as caloric content. Future studies with a similar study design but a larger sample size should be conducted to confirm the current results.

## 5. Conclusions

The present findings indicate that a protein-rich supplementation regime attenuates exercise-induced muscular and inflammatory stress responses and protects against a possibly insufficient daily protein intake by endurance athletes during the training period. The correlations between protein intake and acute exercise-induced stress markers appear to be of relevance for the training of endurance athletes in that the extent of muscle damage and accompanying inflammatory response can be interpreted as a predictor of subsequent delayed-onset muscle soreness and may be attenuated by protein-rich nutrition.

## Figures and Tables

**Figure 1 nutrients-13-02929-f001:**
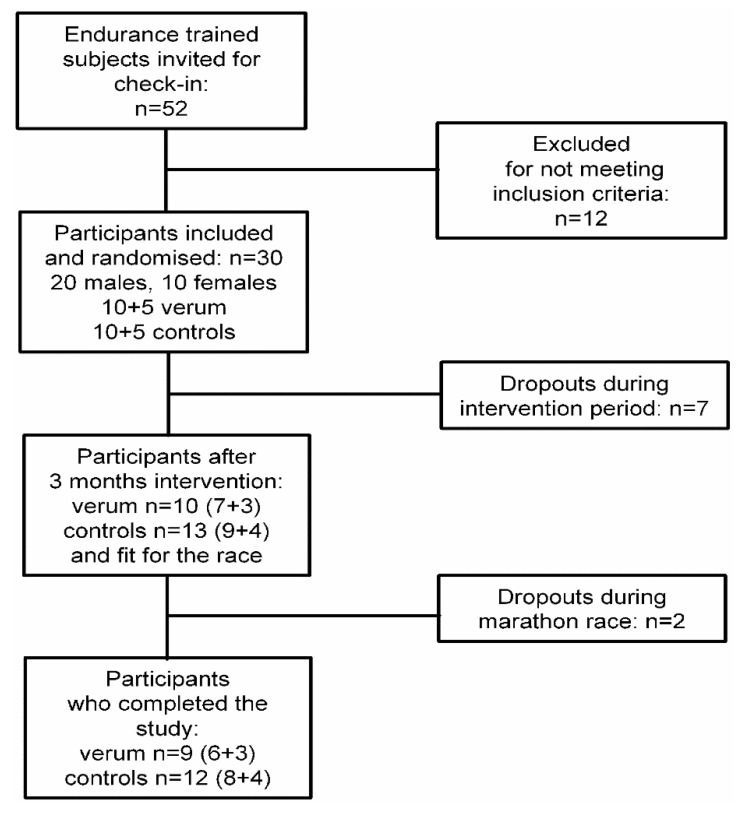
Study flowchart. Control group: endurance training intervention for 3 months. Verum group: endurance training plus protein-rich supplement for three months.

**Figure 2 nutrients-13-02929-f002:**
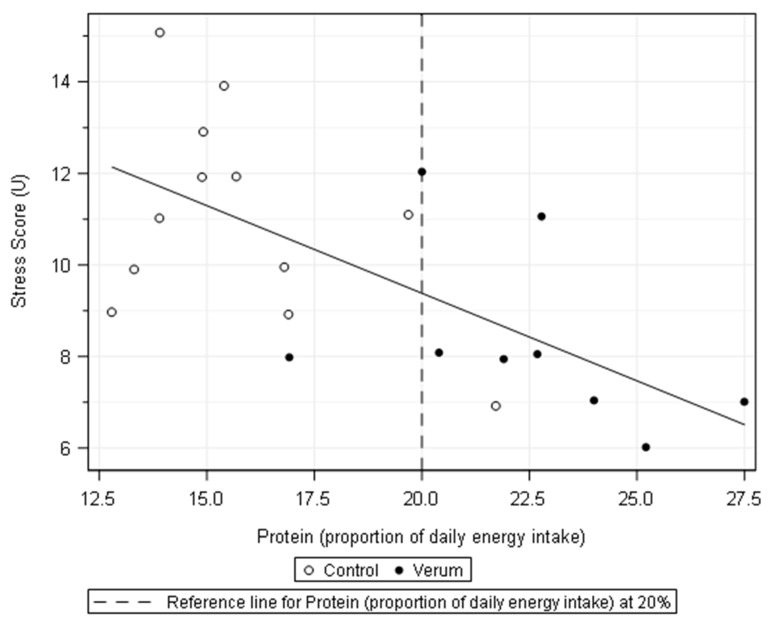
Scatterplot showing a linear regression line for the negative association between the calculated stress score and protein intake (proportion of daily energy intake).

**Table 1 nutrients-13-02929-t001:** Basic and anthropometric characteristics of the study groups (verum, control) before (time point a) and after the 3 months intervention period (time point b).

Variable	Verum (*n* = 10)		Control (*n* = 13)	
	a	b	a	b
Sex (m/f)	7/3		9/4	
Age (y)	29.0 ± 11.0		28.6 ± 8.7	
Body mass (kg)	73.7 ± 6.7	74.5 ± 8.2	71.1 ± 71.5	71.5 ± 8.8
BMI (kg/m^2^)	23.5 ± 1.2	23.3 ± 1.5	22.4 ± 2.1	22.5 ± 1.9
BFM (%BW)	22.9 ± 9.9	21.8 ± 8.9	22.2 ± 7.7	20.9 ± 7.8

Results are presented as mean ± SD. There were no statistical differences between both groups.

**Table 2 nutrients-13-02929-t002:** Performance and exercise-induced stress markers measured before (time point a) and after the intervention (time point b).

Variable	Verum (*n* = 10)		Control (*n* = 13)	
	a	b	a	b
Max. blood lactate concentration (mmol/L)	8.80 ± 1.20	9.60 ± 1.50	9.20 ± 2.30	10.20 ± 3.60
Running speed max (km/h)	16.4 ± 2.0	16.8 ± 1.9	15.9 ± 1.3 *	16.4 ± 1.5
1000 m-time at the IAT (mm:ss)	04:59 ± 00:42	04:44 ± 00:52	05:12 ±00:31	04:41 ± 00:24
VO_2max_ (mL/min/kg)	51.8 ± 6.3 *	53.3 ± 5.9	50.4 ± 4.1 *	51.9 ± 4.9
CK (U/L)	140 ± 120	190 ± 150	170 ± 97	165 ± 95
Myoglobin (μg/L)	35.0 ± 50.0	30.0 ± 15.0	19.0 ± 7.1	18.2 ± 8.0
Leukocytes (1000/μL)	5.3 ± 0.9	5.0 ± 0.3	5.4 ± 1.3	5.2 ± 1.1
IL-6 (pg/mL)	2.0 ± 0.0	2.1 ± 0.2	2.0 ± 0.0	2.1 ± 0.2
Cortisol (ng/mL)	167.00 ± 31.00	206.00 ± 58.00	161.00 ± 36.00 *	195.00 ± 45.00

Results are presented as mean ± SD. * *p* < 0.05; CK, creatine kinase.

**Table 3 nutrients-13-02929-t003:** Exercise-induced stress values of the 21 participants who completed both the intervention and the marathon race before (time point c) and after the race (time point d).

Variable	Verum (*n* = 9)			Control (*n* = 12)		
	c	d	inc. F	c	d	inc. F
CK (U/L)	122 ± 41 **	530 ± 220	4	180 ± 110 **	1400 ± 1300	8
Myoglobin (μg/L)	26.0 ± 15.0 **	770 ± 540 †	29 †	32.0 ± 35.0 **	2200 ± 1800	68
Leukocytes (1.000/μL)	5.3 ± 0.5 **	15.8 ± 2.5	3.0	5.4 ± 1.1 **	18.0 ± 4.2	3.3
IL-6 (pg/mL)	2.0 ± 0.0 **	37.0 ± 30.0	18	2.0 ± 0.1 **	43.0 ± 15.0	21.3
Cortisol (ng/mL)	184 ± 48.0 **	302 ± 70.0	1.6	194 ± 48.0 **	335 ± 95.0	1.7

Results are presented as mean ± SD. ** *p* < 0.01 (intra-group comparisons between time points). † *p* < 0.05 (inter-group comparison at time point and for rise factor). Group-specific increment factor of the measured variables (inc. F.) was calculated as d/c. CK, creatine kinase.

**Table 4 nutrients-13-02929-t004:** Differences in exercise-induced stress markers and the calculated stress score among the athletes who completed the marathon race (*n* = 21) differentiated by their protein.

Variable	PI < 20% (*n* = 12)	PI ≥ 20% (*n* = 9)	r (Diff.y = f(PI%))	Sign. r
Δ CK (U/L)	1.372 ± 1.242	521 ± 238 *	−0.387	0.083
Δ Myoglobin (µg/L)	2.129 ± 1.770	838 ± 602 *	−0.421	0.057
Δ Leukocytes (1.000/mL)	18.180 ± 3.920	15.160 ± 2.540	−0.411	0.064
IL−6 (pg/mL)	40.9 ± 16.2	31.0 ± 17.1	−0.476	0.034
Cortisol (ng/mL)	361 ± 90.4	278 ± 75.6 *	−0.459	0.036
Stress score (U)	11.2 ± 2.12	8.2 ± 1.99 **	−0.667	0.001

Results are presented as mean ± SD. ** *p* < 0.01; * < 0.05 (PI < 20% vs. PI ≥ 20%). Differences in exercise-induced stress markers and the calculated stress score among the athletes who completed the marathon race (*n* = 21) differentiated by their protein intake (PI, % per energy intake per day) after the intervention and the correlation coefficients (protein % of daily energy intake vs. stress markers). CK, creatine kinase.

## Data Availability

The data presented in this study are available on reasonable request from the corresponding author when Almased-Wellness-GmbH gave permission.

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
