# Peer review of "Continuous Protein Supplementation Reduces Acute Exercise-Induced Stress Markers in Athletes Performing Marathon"

_nutrients, 2021, doi:10.3390/nu13092929_

Round 1

Reviewer 1 Report

Review done very well!
I have no further comments to make.

Author Response

The authors would like to thank again the reviewer for his kind engagement with the manuscript. In all cases, the comments contributed were already considered in the revised version of the final paper.

It was important for the authors to emphasise that the importance of carbohydrate intake for endurance performance should not be questioned, but rather the additional value of sufficient protein intake in the training phase of endurance athletes should be demonstrated.

The comments of the second reviewer were also considered in detail.

Reviewer 2 Report

Dear Authors

After check this version, consider all comments in order to improve the final document.

In Advanced

King Regards

Author Response

The authors would like to thank again the reviewer for the intensive and careful engagement with the manuscript. In all cases, the comments contributed were also considered and included accordingly in the revised version of the final paper.

It was important for us to emphasise that the importance of carbohydrate intake for endurance performance should not be questioned, but rather the additional value of sufficient protein intake in the training phase of endurance athletes should be demonstrated.

Here is a summary of our changes:

- all orthographic or linguistic deficits have been corrected

- the introduction was developed

- the references for the running test were included [14, 20] – p.3 / l.125

- the reference for the EBISpro system were included [22 – p.3 / l.143

- all references for the laboratory measurements were included [15-19] / p.3 / l. 107-110

- details to the measurement of the anthropometrics were added (p.3 / l.100-103)

- a calculation for the effect size was not done, before it was an explorative design only

- the information about the ethics were moved to the participants paragraph (p.2 / l.80-85) and the reference of the 7th edition of the Helsinki Declaration was included [12] (p.2 / l.83)

- references about the adaptation to endurance exercise were included [14, 20] (p.9 / l.276

- the references in the conclusion part were removed

- the registration of the study by the university hospital Freiburg were noted (p.9 / l.310)  

Round 2

Reviewer 2 Report

Dear Authors:

After REVIEW THE SECOND VERSION

ACCEPTED IT

CONGRATULATIONS

Author Response

thank you

This manuscript is a resubmission of an earlier submission. The following is a list of the peer review reports and author responses from that submission.

Round 1

Reviewer 1 Report

Although the study is interesting, it seems to me that there are some elements of the design that need to be clarified:

The first thing to note is that it is very difficult to show that the EIMD and stress produced during a marathon are a consequence of the intake or not of a certain protein. We must remember that there is sufficient evidence to show that the food intake of the previous week (Mielgo-Ayuso et al., 2020), the CHO intake during a marathon is of utmost importance to determine the EIMD and stress produced during the same ( Virivay et al. 2020) What would make more sense is to demonstrate whether supplementation with certain proteins allows better recovery during a certain period. For this, it would be sufficient to control the intake and exercise carried out during this period and to assess it in physical / physiological tests and in analytical data. That is, what arises between a and b. Also, the mix of men and women may not be suitable. It has been determined through a meta-analysis (Romero-Parra et al., 2021) that the menstrual phase can influence the EIMD. Therefore, not controlling for this fact in female athletes is a major bias.

Using an equation that determines fat by 4 folds eliminates the leg folds which is not suitable. Better to use Carter.

Taking into account that the intake of the week before a marathon can influence the EIMD, exactly what food groups they ate before the marathon was controlled. Did they all eat the same food groups? If you have a 3-day record prior to the marathon, it might be interesting to include the food groups taken by each group in order to eliminate possible biases.

On the other hand, did any of the participants have specific nutritional control?

In the same way, it has been shown that the consumption of CHO during a mountain marathon can alleviate the EIMD. In this sense, was the CHO and / or water intake of the participants controlled during the marathon?

Although the authors have wanted to unify the sample including as many men as women in both groups, this combination is not appropriate, since the moment of the menstrual cycle can interfere with physiological aspects that may generate a bias.

How was it determined that the participants in both groups performed the marathon at the same intensity? EIMD and internal stress are determined by the intensity and duration of the effort.

In general, although it seems to me that it is a good idea, the method used to control variables especially during the marathon makes it difficult for it to be accepted. I would recommend that the authors review the points I have indicated and try again.

Author Response

Although the study is interesting, it seems to me that there are some elements of the design that need to be clarified:

The first thing to note is that it is very difficult to show that the EIMD and stress produced during a marathon are a consequence of the intake or not of a certain protein. We must remember that there is sufficient evidence to show that the food intake of the previous week (Mielgo-Ayuso et al., 2020), the CHO intake during a marathon is of utmost importance to determine the EIMD and stress produced during the same ( Virivay et al. 2020) What would make more sense is to demonstrate whether supplementation with certain proteins allows better recovery during a certain period. For this, it would be sufficient to control the intake and exercise carried out during this period and to assess it in physical / physiological tests and in analytical data. That is, what arises between a and b. Also, the mix of men and women may not be suitable. It has been determined through a meta-analysis (Romero-Parra et al., 2021) that the menstrual phase can influence the EIMD. Therefore, not controlling for this fact in female athletes is a major bias.

Using an equation that determines fat by 4 folds eliminates the leg folds which is not suitable. Better to use Carter.

Taking into account that the intake of the week before a marathon can influence the EIMD, exactly what food groups they ate before the marathon was controlled. Did they all eat the same food groups? If you have a 3-day record prior to the marathon, it might be interesting to include the food groups taken by each group in order to eliminate possible biases.

On the other hand, did any of the participants have specific nutritional control?

In the same way, it has been shown that the consumption of CHO during a mountain marathon can alleviate the EIMD. In this sense, was the CHO and / or water intake of the participants controlled during the marathon?

Although the authors have wanted to unify the sample including as many men as women in both groups, this combination is not appropriate, since the moment of the menstrual cycle can interfere with physiological aspects that may generate a bias.

How was it determined that the participants in both groups performed the marathon at the same intensity? EIMD and internal stress are determined by the intensity and duration of the effort.

In general, although it seems to me that it is a good idea, the method used to control variables especially during the marathon makes it difficult for it to be accepted. I would recommend that the authors review the points I have indicated and try again.

Reviewer 2 Report

Dear  Authors:

After check the manuscritp some comments are proposed  in order to improve the text. 

King Regards

Author Response

First and foremost, the authors would like to thank the reviewer for the intensive and careful engagement with the manuscript. In all cases, the comments contributed were considered and included accordingly in the revised version of the paper.

Insofar as the comments did not relate to linguistic or orthographic, but to content-related issues or deficiencies, the following explanations are additionally relevant:

- keywords (line 25): The keywords were optimized.

- RCT (lines 179-180): Meanwhile the Nutrients Assistant Editor has received the documents to the inform consent which would be signed by all participants before starting the study as well as the positive vote of the Freiburg University for conducting our study. Please note, that our study was performed as an experimental RCT, but not as a clinical study with the aim of formulating a therapeutic claim. Therefore, it was not necessary to registrate the trial in the DRKS (German Register of Clinical Trials). As official registration number the University vote "231/12" was used.

- marathon race (line 60): The marathon race was now described as part of the intervention.

- inclusion criteria (lines 78-91): The inclusion criteria were listed separately.

- anthropometrics were described in detail (lines 115-120)

- the results of the correlation analysis were given in tab.6; the consideration of data distribution was described in the statistics section (lines 173-176)

- as usual, the note about the registration of the study was given at the end of “methods”; the reference to the Helsinki Declaration is also listed here (lines 177-183)

- there are no statistical differences between the two groups (inter-group comparison) at baseline (tab. 1-4)

- in the first part of the introduction the main aims of the study have been presented (lines 279-282)

- the reference to the “health benefit” were included as [24] (line 298)

- the recently published references on the importance of carbohydrates in endurance sports were included [45-47] (lines 340-343)

Reviewer 3 Report

  1. It's a well-designed study, however, the lack of dietary records or relative data may affect the results. especially the data showed that protein proportional intake has negative correlation with stress score.
  2. if the data analysis of dietary records can be showed, it will give us some evidence and more clues to tell if protein intake is really an important factor of stress markers in endurance athletes. 
  3. It is recommend that the author should discuss the effect of other supplement formula on exercise-induced stress, for example, coenzyme Q10 and carnitine, etc.

Author Response

Comments to review-3

The authors would like to thank the reviewer for the evaluation and kind feedback on the manuscript. The comments were included in the revision of the paper.

As recommended, the possible effects of ergogenic substances, such as coenzyme Q10 and carnitine, on endurance performance and exercise-induced stress were discussed according to the recommendations of the ACSM [1] (lines .331-340).

Reviewer 4 Report

The topic is very interesting. The manuscript is well written and methodologically correct. 

The authors investigated changes in endurance performance and metabolic, hormonal, and inflammatory markers induced by endurance stress (marathon race) in a combined strategy of training and dietary protein supplementation.

Results showed that protein intake and protein-related nutrients (20% of total energy intake) may influence exercise-induced muscle stress reactions by changing cellular metabolism and inflammatory pathways. 

I only have a few minor recommendations to make to the authors:

Keywords

To optimize the search of the manuscript on the search engines, insert different keywords from those present in the title. Remove “body composition” and “endurance performance” and, if necessary, add other relevant key words. It is important to add words other than those present in the title.

Materials and Methods

For ease of reading and understanding the research methods, in the Materials and Methods section, I recommend sub-dividing the contents with sub-sections: Study design, Participants, Procedures and Measures, and Statistical Analysis.

Line 173: Replace with "Results".

Figure 1 is not well readable.

Author Response

Comments to Review-4

The authors would like to thank the reviewer for the evaluation and kind feedback on the manuscript. The comments have been  included in the revision of the paper.

As recommended

- the keywords were optimized (lines 25-26)

- the method section has been divided into sub-sections

- “Result” was replaced by “Results” (line 184)

- the text of fig. 1 was reformulated for better understanding (272-273)

Round 2

Reviewer 1 Report

The manuscript still has the same problems as in the first round. Therefore, although the idea is good, I think that until you solve the problems it would not be suitable for publication.

Author Response

As already stated in my cover letter to the first round of reviewers, the conceptions and the proposed changes of reviewer no. 1 can only be implemented to a limited extent. In my detailed commentary on the corresponding review, I have tried to take the proposed changes into account as far as possible and to incorporate them into the manuscript.

Reviewer 2 Report

Dear Authors:

After check the second version, minor details are presented to consider

1) PAGE 194, 3 MONTHS BETTER

2) PAGE 230, VO2 MAX, 2 MAX in subindex

3) PAGE 233, Subjects, better participants

Author Response

Changes done.

Reviewer 4 Report

Thanks for the changes made!

Author Response

Thanks.